# Characterization of the Maximum Height of a Surface Texture

**DOI:** 10.3390/ma16227109

**Published:** 2023-11-09

**Authors:** Pawel Pawlus, Rafal Reizer, Wieslaw Żelasko

**Affiliations:** 1Faculty of Mechanical Engineering and Aeronautics, Rzeszow University of Technology, Powstancow Warszawy 8 Street, 35-959 Rzeszow, Poland; 2Institute of Materials Engineering, College of Natural Science, University of Rzeszow, Pigonia Street 1, 35-310 Rzeszow, Poland; rreizer@ur.edu.pl; 3Faculty of Mechanics and Technology, Rzeszow University of Technology, Kwiatkowskiego Street 4, 37-450 Stalowa Wola, Poland; w.zelasko@prz.edu.pl

**Keywords:** surface texture, profile, parameters, stylus profilometer, white light interferometer

## Abstract

Average surface height and maximum amplitude can affect surface functions. In the industry, these parameters can be obtained based on profile measurements. However, variability in maximum profile height is high. A more stable parameter can be obtained from the results of the areal surface topography measurements as the average value of the parallel profiles. The aim of this study is to establish this parameter directly from the result of the areal surface texture by correcting the maximum surface height to material ratios in the range of 0.13–99.87%. This method was tested by measuring 100 surface topographies with a stylus profilometer and a white light interferometer. It can be utilized correctly for deterministic textures and random one- and two-process surfaces for which the correlation between neighboring profile ordinates is not very high. In other cases, the method should be modified. Employing this method, the maximum profile amplitude Pt and parameters characterizing the average profile height Pq, Pa, and the ratios Pq/Pa and Pp/Pt describing the shape of the profile ordinate distribution can be correctly estimated. Pq/Pa and Pp/Pt were more stable than the kurtosis Pku and skewness Psk. The corrected maximum height S_±3σ_ can be adopted as a parameter that characterizes the areal surface texture as more stable than the maximum surface height St. Pq/Pa and Pp/Pt were more steady than kurtosis Pku and skewness Psk.

## 1. Introduction

Surface texture affects the functional properties of machine elements, including contact, sealing, friction, and wear [1,2,3]. Measurement and analysis of areal surface topography can help correct the assessment of surface properties, and profile analysis can lead to false surface characterization [4]. Measurement of areal surface topography is now very popular. Previously, the stylus technique was utilized. Errors typical for the stylus technique are connected to the contact of the stylus tip with the measured surface [5,6,7]. Optical methods replaced the stylus technique in areal surface topography measurement due to their much shorter measurement time. However, optical methods are sensitive to measurement errors. Errors typical for optical methods are mainly caused by the presence of spikes and non-measured points [8,9,10]. Spikes are narrow peaks that do not occur on a real surface. Non-measured points are surface locations for which no valid measured values occur. The application of stitching is an additional source of measurement error [11,12,13].

Analysis of the topography of the areal surface can add new information compared to profile analysis; therefore, many new parameters have been introduced, such as feature parameters characterizing the peaks [14,15]. The amplitude parameters are believed to mostly affect the surface properties, for example, they are related to friction and wear. The opinion exists that maximum height is related to surface damage while the averaged parameters are related to surface normal functioning. The maximum roughness height can be employed to detect cracks in the surface layer.

In industrial applications, profile analysis is most commonly utilized for the assessment of the machining process, and two amplitude parameters are typically applied: the arithmetical roughness mean height Ra and maximum heights Rz or Rt (Rz is calculated on the sampling length, whereas Rt is calculated on the evaluating length; the sampling length is smaller). These parameters are employed to characterize surface roughness in most engineering applications because of their substantial functional importance and ease of use.

Topographic surface analyses should be related to profile studies. Therefore, the amplitude parameters of surface topography are extensions of roughness profile parameters. In the measurement of areal surface texture, the rms height Sq is preferred for characterizing the mean surface amplitude over Sa because of its greater statistical character. However, the Sa parameter is also utilized because the Ra parameter is still very popular in the industry. Sa and Sq are similar to Pa and Pq (unfiltered profile parameters), respectively. The maximum surface amplitude is characterized by St (or Sz). However, the St parameter is significantly larger than the Pt parameter [14,16], particularly for random surfaces. This is related to the higher number of data points in areal surface analysis compared to profile analysis [17,18]. The maximum amplitude of the areal surface texture St can lead to false results. For example, the presence of spikes after surface texture measurement adopting optical methods overestimates the surface height.

The maximum profile height Pt is not a stable parameter [19]. This fluctuates between different surface realizations or measurements. Maximum profile height Pt is also variable. High variability of maximum profile height can lead to false qualifications of well-done surfaces (as shortages). Persson [17] proposed adopting a more stable parameter than maximum profile height calculated in the usual manner by averaging the maximum profile heights over all scan lines of the measured areal surface topography. Another possibility for eliminating the instability of the maximum profile height is correction. Zhang et al. [20] restricted the probability distribution to 95% and 99% for calculating the Rz and Rt parameters, respectively. This procedure reduces the effect of noise on the surface roughness created in ultra-precision machining. Guo et al. [21] applied this concept and reduced the boundary effect to a 95% interval for the areal surface texture after diamond turning.

Skewness Rsk and kurtosis Rku characterize the shape of the ordinate distribution of the roughness profile. Ssk and Sku are the areal extensions of these parameters. Rsk and Ssk are similar, whereas Sku is typically larger than Rku. Pawlus et al. [22] found that Rp/Rt and Rq/Ra are alternatives to Rsk and Rku, respectively. They described the two-process textures better, and the Rq/Ra ratio was more stable than Rku. There are problems with establishing a stable maximum surface texture height. This problem is tribologically important [13]. Persson’s proposal [17] to increase parameter stability by calculating the average value of the maximum profile heights over scan lines is interesting. This study attempts to establish a parameter similar to the mean maximum height of all parallel profiles based on areal surface texture measurements. This problem is very important because maximum and average surface heights are commonly used in most practical applications. The new parameter is more reproducible than maximum profile height calculated in the usual manner.

Another objective is to compare the stability of the profile parameters characterizing the shape of the ordinate distributions Pp/Pt and Pq/Pa over Psk and Pku.

## 2. Idea of Parameter Calculation

The height distribution of the random one-process surface texture (with traces from one process) is typically Gaussian. Surface texture is treated as a random variable. The scatter of random variables depends on the sample size, which is the number of measuring (data) points. The probability of the presence of any point outside the given range of the Gaussian ordinate distribution can be calculated. The general form of the probability density function of the Gaussian distribution is:(1)$$f\left(x,\;\mu ,\;\sigma \right)=\frac{1}{\sigma \sqrt{2\pi }}e^{-\frac{1}{2}{\left(\frac{x-\mu }{\sigma }\right)}^{2}}$$
where *µ* is the mean value and *σ* is the standard deviation. For a given range of input variables *x*_1_ and *x*_2_, we can compute the inner cumulative distribution, which is the probability that the variable can be between *x*_1_ and *x*_2_:(2)$$P\left(x_{1}\sim x_{2},\;\mu ,\;\sigma \right)=\int_{x_{1}}^{x_{2}}f\left(x,\;\mu ,\;\sigma \right)dx$$

The outer cumulative distribution complements the inner cumulative distribution. This is the probability that the random variable is outside the range of *x*_1_ to *x*_2_:(3)$$Q\left(x_{1}\sim x_{2},\;\mu ,\;\sigma \right)=1-P\left(x_{1}\sim x_{2},\;\mu ,\;\sigma \right)$$

It was assumed that the maximum surface amplitude was at a height outside of which only two points occurred. For 1000 points, the probability is 0.002. This probability corresponds to the range ±3.09σ (Figure 1)—outer cumulative *Q* = 0.002. Similarly, one can predict that the maximum corresponding to 2000 points is in the range of ±3.2σ, 200,000 points ±4.5σ, and 1,000,000 points ±4.7σ. σ equals the profile Pq parameter and areal surface texture parameter Sq. For the surface profile of Gaussian ordinate distribution comprising 1000 points, the maximum profile height Pt (or Rt of the roughness profile after eliminating long wavelengths) should be approximately 6.2 Pq (or Rq). Seewig [23] obtained similar results. He found that for 1000 points of the Gaussian roughness profile, the Rt parameter was equal to the Ra parameter magnified by eight. For the Gaussian profiles, the ratio Rq/Ra is 1.25; therefore, Rt equals 6.4 Rq. This value is similar to that obtained here.

It was assumed that for approximately 1000 ordinates, the maximum profile height was equal to six standard deviations (Pq parameter). The amplitude of the areal surface texture can be corrected using a material ratio (Abbott–Firestone) curve [24]. It describes the functional properties of machined parts, such as wear intensity and load-carrying capability. It can be utilized to characterize the low wear of machine elements (wear within the limits of the original surface topography) and oil capacity (volume of oil maintained within the surface texture). The Abbott–Firestone curve is frequently employed for production control. This curve presents the cumulative surface height distribution (see Figure 2a). It can be adopted for 2D profile and 3D areal surface texture analyses. The surface amplitude corresponding to six standard deviations can be obtained by truncating the surface height of the areal surface texture corresponding to material ratios of 0.13 (peak part) and 99.87% (valley part), known as surface thresholding. This action is easy. This parameter is called S_±3σ_. Figure 2b presents the scheme of obtaining S_±3σ_. This should equal the average value of the maximum heights (Pt or Pz) of all profiles from the measured surface. Figure 2c presents a probability plot of the material ratio curve in a Laplace-normal system (probability curve). The standard deviation is a unit of the material ratio. The height corresponding to the standard deviation of the Gaussian surface is equal to σ (the Pq parameter of the profile or the Sq parameter of the surface). A straight line is visible for the random texture of the Gaussian ordinate distribution. The slope of this curve is equal to σ. The profile height is equal to 6σ (±3σ), and the surface height is equal to 9σ (±4.5σ).

One can presume that the S_±3σ_ parameter can be applied to characterize only surfaces of Gaussian ordinate distribution. However, it can also be utilized for bi-Gaussian and multi-Gaussian surfaces. Bi-Gaussian surfaces are two-process surfaces with traces from two processes. Plateau-honed surfaces fabricated from cylinder liners are the most popular examples of two-process random surfaces. They are created via two processes: finish honing and plateau honing. Plateau-honed cylinder structures have advantages over those obtained after one-process honing, characterized by the same average surface height, lower friction, wear, and tendency to seizure. The one-process random surfaces after mild wear are also two-process surfaces. The probability plot of the bi-Gaussian surfaces exhibits two straight lines [24,25,26,27]. The upper line characterizes the plateau part, whereas the lower line describes the valley part. The slope of the upper straight line is equal to Ppq (Spq) (the standard deviation of the plateau part), whereas the slope of the lower straight line is equal to Pvq (Svq) (the standard deviation of the valley part). The Pmq (Smq) parameter is the material ratio at the transition point from the plateau to the valley portions. Because bi-Gaussian surfaces are created by the superimposition of two Gaussian structures, the S_±3σ_ parameter can be obtained for bi-Gaussian surfaces and, generally, for multi-Gaussian textures.

The presented analysis and the work of Seewig [23] were conducted assuming that the data points were not correlated. Whitehouse and Archard [28] analyzed uncorrelated profile ordinates because only long wavelengths existed in the surface textures after wear. They proposed that the sampling interval should equal the correlation length CL (the distance at which the autocorrelation function decays to 0.1). However, in practice, the sampling intervals are considerably smaller than the correlation length. The minimum sampling interval is restricted, depending on the measurement method. For example, when employing the tactile method, the sampling interval should not be smaller than the radius of the stylus tip. In the optical measurements, the small peaks or valleys inside the unit beam spot on the specimen cannot be detected. The method presented above for determining the average value of the maximum profile height should be modified when the sampling interval is much smaller than the correlation length, that is, when the ordinates of the neighboring points are highly correlated. The profiles of the Gaussian ordinate distribution were modeled by adopting the procedure developed by Wu [29]. Each profile, with a 2 mm length, contained 1000 data points.

The height of each profile characterized by the Pq parameter was 1 µm. The correlation length CL changed from 2 to 100 µm. It was found that an increase in the correlation length for the same sampling interval (2 µm) caused an increase in the PSm parameter (mean width of the elements of the profile) and a decrease in the rms slope Pdq. Interesting changes were observed in the probability plot of the Abbott–Firestone curve. The horizontal distance at which Pq was correctly estimated, based on the profile probability plot (straight line), decreased as the correlation length of the profile increased. Figure 3 presents an example of this model. This distance is between the two vertical dotted lines. When the correlation length CL was equal to 4 µm, this distance corresponded to a height of 6σ (±3σ). When the correlation length CL increased to 60 µm, this distance corresponded to a height of 5.2σ (±2.6σ). Therefore, for a higher correlation between neighboring ordinates, the corrected surface height should be modified (smaller).

The above-presented analysis including Equations (1)–(3) is applicable for random surfaces. The authors of this paper test if the maximum height correction of areal surface texture measurement to S_±3σ_ can also be adopted for nominally periodic surfaces after milling and turning. A larger height correction has been previously employed [20,21] to eliminate random components.

## 3. Material and Methods

One hundred surface textures were measured with the optical (sixty surfaces) and stylus (forty surfaces) methods. These two approaches use different techniques. There are various sources of measurement errors after applications of these methods [5,6,7,8,9,10,11,12,13]. They were utilized to prove that our approach can be used independently of measurement methods. One-process steel surfaces were machined with various methods, including polishing, lapping, grinding, vapor blasting, turning, and milling. After turning and milling, the surfaces were periodic, and the other surfaces had random characteristics. The discs subjected to polishing, lapping, grinding, milling, turning, and vapor blasting were made of 42CrMo4 steel of 40 HRC hardness obtained after heat treatment. Surface topographies of gray cast iron after one-process honing and plateau honing were also measured and analyzed. After the plateau honing, the surfaces were two-process bi-Gaussian. Honed cylinder liners were made from gray cast iron of 220–260 HB hardness. Surfaces after polishing, lapping, grinding, and milling were flat; discs of typically 25.4 mm diameter were utilized. Diameters of cylindrical samples after turning and honing were in the range 80–130 mm. Surface textures with low content of waviness were analyzed.

The surface textures were measured with a white light interferometer (Talysurf CCI Lite with 0.01 nm vertical resolution). In these cases, the measurement conditions were the same: the measuring area 3.29 mm × 3.29 mm contained 1024 points (objective 5× was utilized). The sampling intervals were 3.22 µm in perpendicular directions.

Surface topographies were also measured with a stylus profilometer Form Talysurf i-series equipped with a tip radius of 2 µm; the stylus force was 1 mN. Each surface was measured at a speed of 0.5 mm/s. During profile measurement in the x direction, movement of the stylus tip occurred, and the table with the sample was stationary. The table was moved perpendicular to the stylus tip to measure the areal (3D) surface topography. The initial sampling intervals were 0.1 µm in the measurement direction x and 5 µm in the perpendicular direction. Before analysis of the measurement results, the sampling interval in the x direction increased to 2 or 3 µm, depending on the assessment length. This selection depended on the size of the measuring sensor. The sampling interval should be not smaller than the tip radius [1,5,30,31]. When the optical method was applied, the ratio of non-measuring points was smaller than 5%, and the non-measured points were filled in. In each case, the number of data points in the measurement direction was close to 1000. The numbers of data points and assessment lengths perpendicular to the x direction were various; the total number of points varied between 200,000 and 1,000,000. The post-processing procedure was the same and independent of the measurement method employed. The flat surfaces were leveled without digital filtration. The forms of the curved surfaces (after honing and plateau honing) were removed by employing the second-level polynomials; digital filtration was also not utilized. An attempt was made to analyze only the surface topographies with a small number of spikes (surfaces with many spikes were eliminated) when the optical method was applied.

The x direction in which the profile parameters were calculated for the anisotropic one-directional surfaces (after grinding, milling, and turning) was perpendicular to the main direction of the texture (lay). This direction corresponds to a smaller correlation length (also the PSm parameter) from both perpendicular directions for the honed plateau-honed, cross-hatched, and other surfaces. All profiles of the surface textures in the x direction were determined, and the average values of the following parameters were calculated: arithmetic average roughness height, Pa, rms height Pq, skewness Psk, kurtosis Pku, peak height Pp, and maximum height Pt (or Pz). The Pq/Pa and Pp/Pt (emptiness coefficient) ratios were also calculated. Then, autocorrelation functions were determined for a few representative profiles and from these functions, the correlation lengths CL were obtained. The number of profiles (typically five was enough), for which the autocorrelation functions were determined, depended on surface character; it should be higher for nonhomogeneous surfaces. The analysis of profile autocorrelation functions was not performed for nominally periodic surface textures (after milling and turning). The maximum heights of the areal surface textures were corrected based on the correlation lengths. In this work, unfiltered parameters of corrected areal surface texture were compared with their substitutes of series of unfiltered profiles. However, in-industry roughness profiles are commonly analyzed. Therefore, before industrial application of this method, filtering should be taken into account.

The x direction corresponds to smaller correlation length in perpendicular directions of anisotropic surfaces. For random surfaces with a correlation length CL not larger than 50 µm in the x direction (or smaller than the horizontal distance between the 20 measuring points), and for surfaces after turning and milling, the maximum amplitude was corrected to heights corresponding to material ratios between 0.13 and 99.87%. This corrected height was called S_±3σ_. When the correlation length of the random surfaces was greater than 50 µm, the maximum areal surface amplitude was corrected to a height corresponding to material ratios between 0.3 and 99.7%. This modified corrected height is denoted as S_±2.75σ_.

This research found a CL correlation length greater than 50 µm for some surfaces after vapor blasting; the maximum correlation length was 75 µm. For a higher correlation length, further modification of the corrected maximum surface height should be performed (range of surface height should be reduced).

The following parameters were calculated for surface textures with corrected maximum heights: average surface height Sa, rms surface height Sq, peak height Sp, maximum height St or Sz (equal to the corrected height), skewness Ssk, and kurtosis Sku. Similar to the profiles, the ratios Sq/Sa and Sp/St (emptiness coefficient) were calculated. Finally, the absolute and relative errors of the values of Sa, Sq, St, Ssk, Sku, Sp/St, and Sq/Sa of the areal surface textures with the corrected maximum height compared with the mean values of the profile parameters Pa, Pq, Pt, Psk, Pku, Pp/Pt, and Pq/Pt were calculated, and the mean profile parameters were utilized as reference values.

Figure 4 presents a procedure of surface texture study after measurement and initial analysis (leveling of flat surfaces or form removal of cylindrical samples, filling-in of non-measured points after optical measurement, change in sampling interval after measurement using the stylus technique).

## 4. Results and Discussion

Table 1 presents the accuracy results of the estimated average values of the profile parameters in the x direction, based on the measurement of the areal surface texture.

The errors of profile parameter estimation Δ were calculated as:(4)$$\Delta =\left|\frac{S_{par.}-P_{par.}}{P_{par.}}\right|$$
where *S_par._* is the parameter of areal surface texture with corrected maximum height, and *P_par._* is the mean value of the profile parameter over all scan lines.

Mean and maximum errors obtained after the analysis of 100 surface textures are listed in Table 1. In addition to the relative errors in estimating the Pt parameter given in Table 1, the errors of Pq, Pa, Psk, Pku, Pp/Pt, and Pq/Pt are presented.

The estimation errors of the profile parameters were found to be independent of the method utilized to measure the areal surface texture. Similar errors were obtained with a stylus profilometer and white light interferometer. In 60% of the analyzed cases, the corrected maximum surface height was higher than the mean value of the Pt parameter.

The mean accuracy of the estimation of the Pt parameter of the surfaces after grinding based on the areal surface topography measurements was 2.3%, and the maximum error was 5.3%. This precision seems good because the surface textures after grinding did not have a Gaussian ordinate distribution—the skewness values were negative (not smaller than −0.5), a characteristic feature of the ground textures [2]. The accuracy of the mean value of the maximum profile height estimation for very smooth surfaces after polishing and lapping was very high, with a mean error of 1.3% and a maximum error of 3.1%. Most surfaces after lapping and polishing had Gaussian ordinate distribution which is probably related to the good performance of the proposed method. The accuracy of the maximum height parameter estimation after one-step honing was poorer, with a mean error of 3.1% and a maximum error of 7.9%. However, in most cases, the ordinate distribution was not exactly Gaussian (similarly to ground surfaces)—the values of the Ssk parameter were negative, not smaller than −0.8. According to Whitehouse [1], one-process Gaussian surfaces can be characterized by skewness between −1 and 1. One-step honed cylinder surfaces with various honing angles were measured and analyzed. When the honing angle was less than 60°, the x direction was axial, and when the honing angle was 125° the x direction was circumferential. Two-process honed surfaces were characterized by a honing angle of about 50°, and the x direction was axial. The Ssk parameter was between −4 and −1.4. The precision of the estimation of the Pt parameter based on areal surface measurements was comparatively high, with a mean error of 1.5% and a maximum error of 3.9%. Among the surfaces after vapor blasting, the modified corrected surface maximum height S_±2.75σ_ was employed in 50% of the analyzed cases. The analysis of the autocorrelation function yielded satisfactory results. Typically, a higher surface amplitude corresponds to a longer CL correlation length. However, the precision of the maximum profile height estimation was not very high, with a mean error of 3.9% and a maximum error of 9.7%. The proposed method of estimating the maximum profile height based on the areal surface texture was also utilized for deterministic surfaces after turning and milling, with a mean error of 3.3% and a maximum error of 7.9%. The noise was eliminated by correcting the maximum height of the areal surface texture to S_±3σ_.

This method performed well for bi-Gaussian surfaces after plateau honing. The performances of this method for other random surfaces with negative skewness values were poorer, which is probably related to the non-Gaussian characteristics of these surfaces. The precision of this method can be improved with careful selection of the measured area.

Figure 5 presents examples of the estimation of the mean values of the selected profile parameters based on the measurement results of areal surface texture. Both surfaces were measured with a stylus profilometer. The sampling intervals were 3 and 2 µm, respectively, for surfaces after grinding and vapor blasting. For the ground surface texture, the maximum profile height estimation error is approximately 1%. The values of Pa and Pq were estimated correctly. The error in the estimation of the emptiness coefficient Pp/Pt was less than 3%. The relative errors of the Psk and Pku parameter estimations were higher, amounting to 7 and 4.6%, respectively. The modified surface correction S_±2.75σ_ was adopted for the surface after vapor blasting because the correlation lengths in the horizontal profiles were approximately 55 µm. The mean values of the maximum heights of the parallel profiles were estimated with a deviation of <2%. The errors in the determination of Pa and Pq were approximately 1%. The Pq/Pt ratio was accurately estimated. The emptiness coefficient Pp/Pt was estimated to have an error of approximately 2%. The relative errors in the estimations of Psk and Pku were higher, amounting to 17 and 7%, respectively.

Figure 6 presents the estimation results of the profile parameters of the surfaces after plateau honing and milling. The surface after plateau honing was characterized by a negative skewness Ssk and a high kurtosis Sku. The x direction was circumferential. The average value of the maximum profile height Pt was determined to have an error of less than 2%. Parameters Pa, Pq, and Pp/Pt were estimated correctly. The errors in the skewness and kurtosis were also small. The error in the estimation of the average value of the maximum height Pt of the series of profiles of the milled surface was 2.5%. The parameters Pq, Pa, and Pp/Pt were correctly estimated. The skewness PSk and kurtosis Pku values were accurately estimated.

The proposed method for estimating the average value of the maximum height of a series of parallel profiles generally yielded good results (Table 1, Figure 5 and Figure 6). The mean error of the Pt parameter estimation of approximately 3% is low compared with variations of profile parameters on typical manufactured surfaces [19]. This method can be utilized for random surfaces of a Gaussian ordinate distribution and bi-Gaussian random textures. It has also been adopted for nominal deterministic surfaces after turning and milling. In this case, the correction of the maximum height eliminated noise. The method of maximum height correction to S_±3σ_ should be modified for a very high correlation between neighboring ordinates of random surfaces. This modification, depending on the correction to S_±2.75σ_, was applied only to some surfaces after vapor blasting and ensured good results. In industrial applications, correction to S_±2.75σ_ should be seldom used because its necessity was found only for very rough surfaces; machined surfaces are typically smooth.

This method was restricted to surfaces, for which the profile correlation lengths were smaller than 75 µm or smaller than the horizontal distance between the 30 measuring points. For greater correlation lengths, further modification of the procedure is needed (range of surface height should be narrowed). However, in scientific works, high correlation should be avoided [28]. This method also allowed for very good estimations of Pq and Pa mean profile parameters. The mean values of the parameters that describe the shape of the profile ordinate distribution, Pp/Pt and Pq/Pa, were also accurately estimated. These results confirm the findings in [22] that these parameters are more robust than Psk and Pku, respectively. However, the accuracy of the kurtosis Pku estimation was also comparatively good. The high relative errors in the skewness Psk estimations were partly caused by the values of this parameter being close to zero, which occurred for surfaces with a Gaussian ordinate distribution. Therefore, in Figure 4 and Figure 5, the surface textures of the Psk/Ssk parameter smaller than −0.5 were analyzed.

This method was tested on profiles containing approximately 1000 points. However, it can also be applied to profiles with 2000 points. When the ordinates of these points are independent (non-correlated), the profile height should correspond to 6.4σ. In areal surface texture measurement, typically employing optical methods, the number of measuring points in perpendicular directions is rarely higher than 2000; otherwise, the stitching procedure should be adopted. Stitching led to the possibility of an increase in measurement area with good horizontal resolution. When the number of measuring points in the x direction is much smaller than 1000 and higher than 2000, the correction of the maximum height of areal texture should be modified. Without correcting the maximum texture height one cannot estimate correctly the maximum profile height and other profile parameters based on the parameters of areal surface texture.

Owing to the short measurement time, the measurement of surface topography employing optical methods is now common. However, the optical methods are highly sensitive to measurement errors [1]. The presence of spikes may be a source of these errors. The presence of spikes causes overestimation and an increase in variations of height parameters. Truncation of surface texture height is the simplest method to remove spikes. The effect of spikes can be also mitigated using other methods such as limitation of slope. This study analyzed the results of areal surface texture measurements without spikes utilizing a white light interferometer. The presence of spikes affected the mean values of the parameters of parallel surface profiles. This effect could be reduced by correcting the maximum height of the areal surface texture. In most cases, the estimated values of the parameters Pq, Pa, Pz, Psk, and Pp/Pt obtained with the correction of the maximum areal surface amplitude were closer to the mean values of these parameters after the elimination of spikes than the average values of these parameters containing spikes.

Because this method performed well for plateau-honed surfaces and deterministic surfaces after milling and turning, it would be probably useful for complex surfaces, such as textured surfaces [32,33,34]. This work focuses on surface texture analysis in the context of metals; however, the proposed method is probably applicable to other types of manufacturing materials, such as ceramics and composites.

Based on the analysis of parameter variation on typical manufactured surfaces, it was also found that the corrected maximum height S_±3σ_ was more stable than the maximum surface height St (or Sz). Therefore, S_±3σ_ can be adopted as an additional height parameter for the areal surface texture. If this parameter is utilized, impacts of non-statistical peaks or valleys will be minimized.

The measurement of areal surface texture can be used in industry for production control and process optimization to obtain stable values of average and maximum height parameters. Based on surface type, the conditions of measurement and analysis such as measuring area and sampling interval should be determined. Filtering should be taken into account.

## 5. Conclusions

The average value of the maximum profile height of the series of perpendicular profiles can be correctly estimated based on the amplitude parameter of the areal (3D) surface texture. The method depends on the correction of the maximum texture height to the S_±3σ_ corresponding to material ratios between 0.13 and 99.87%. This method can be employed for deterministic textures and random one-process and two-process surfaces of profile correlation lengths that are not greater than the horizontal distance between the 20 measuring points. This correction should be modified to include longer correlation lengths to the S_±2.7σ_ corresponding to material ratios between 0.3 and 99.7%; the necessity of this modification was found for 50% of surfaces after vapor blasting.The mean error of Pt parameter estimation was approximately 3%, which was not higher than 10%. Among the tested surfaces, the accuracy of the mean value of the maximum profile height estimation for very smooth surfaces after polishing and lapping was very high. This method behaved well for bi-Gaussian surfaces after plateau honing. Its performances for other random surfaces with negative skewness values were poorer.The proposed method allowed us to estimate the average values of profile amplitude parameters and parameters describing the shape of the amplitude distribution precisely. The mean errors of Pa, Pq, and Pq/Pa were approximately 1%, whereas that of the emptiness coefficient Pp/Pt was approximately 3%. In most cases, the errors in parameter estimations were less than 10%.The results confirmed that Pq/Pa and Pp/Pt were more robust parameters characterizing the shape of the profile ordinate distribution rather than skewness Psk and kurtosis Pku.This method was tested for areal surface textures measured with the stylus method and a white light interferometer for profiles containing approximately 1000 measuring points. It can also be utilized for up to 2000 profile points.The S_±3σ_ parameter can be utilized to characterize the maximum amplitude of the surface texture. The variation in this parameter for the typical manufactured surfaces was smaller than that of St (Sz). This parameter can be useful in scientific works and production control.In further research, the effect of high-frequency filtering on the accuracy of the proposed method will be studied. This method will be applied to other surfaces, including complex ones made from various materials.

## Figures and Tables

**Figure 1 materials-16-07109-f001:**
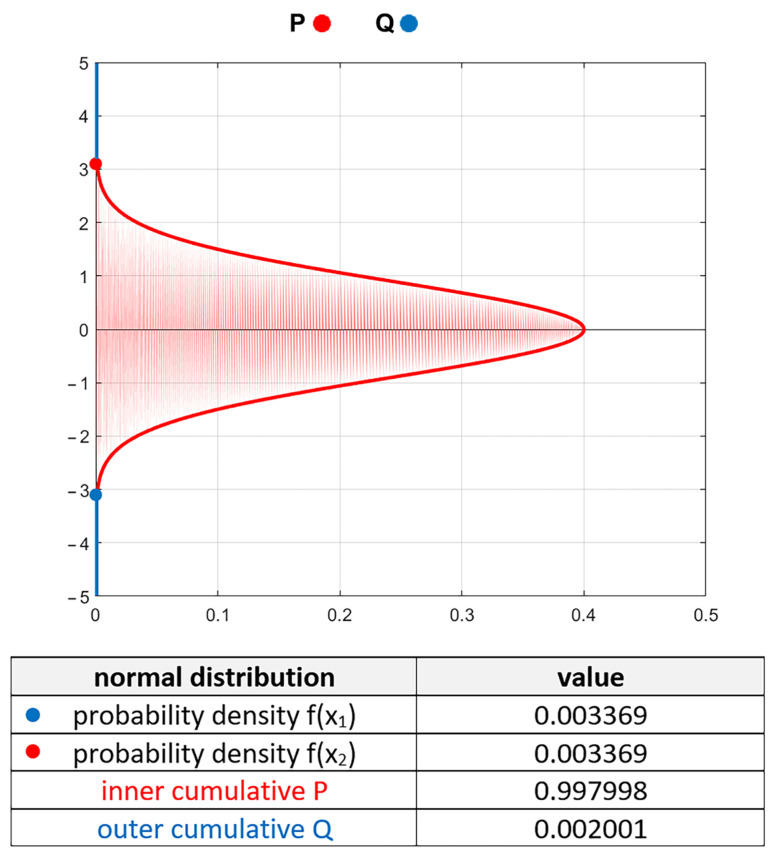
Probability density function of random variable.

**Figure 2 materials-16-07109-f002:**
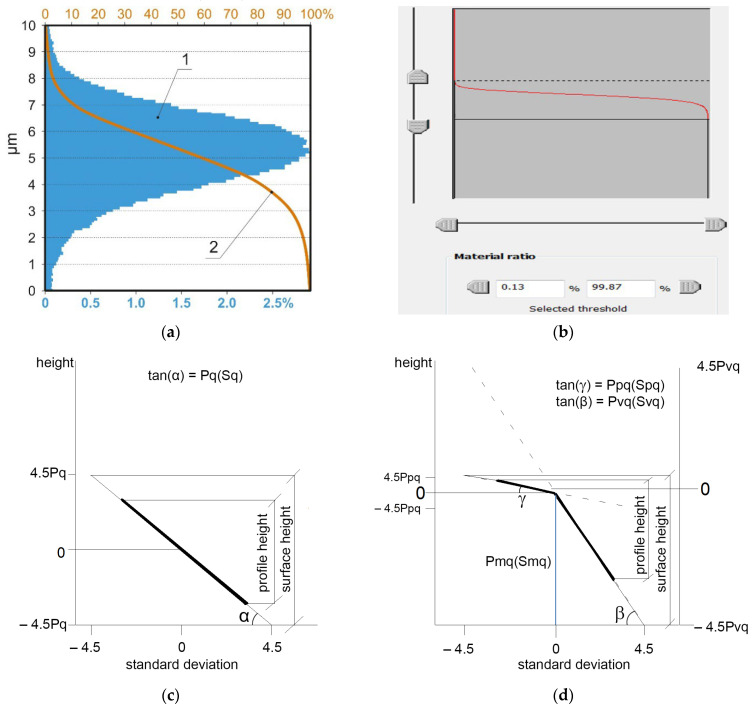
Height distribution 1 and material ratio curve 2 (**a**), scheme of obtaining S_±3σ_ parameter employing material ratio curve (**b**), probability plot of material ratio curve of one-process surface of Gaussian ordinate distribution (**c**), probability plot of two-process bi-Gaussian surface (**d**).

**Figure 3 materials-16-07109-f003:**
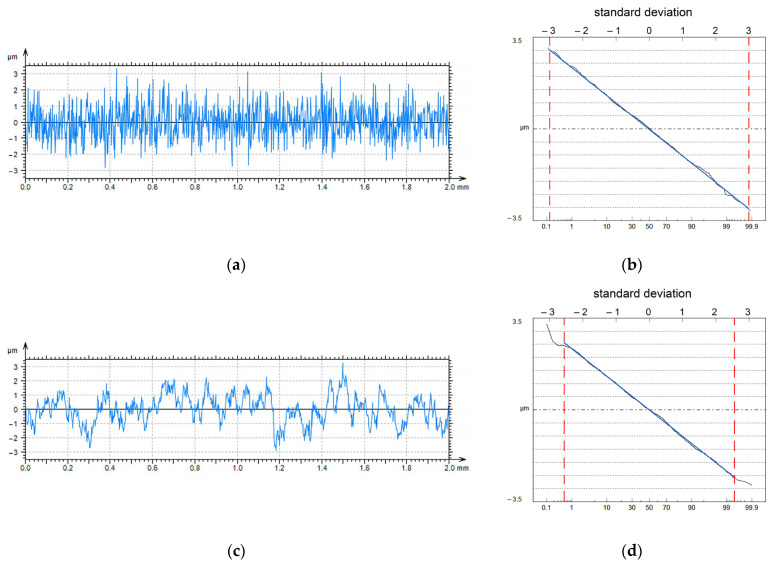
Simulated profile characterized by correlation length CL of 4 µm (**a**), its probability plot (**b**), simulated profile characterized by correlation length CL of 60 µm (**c**), its probability plot (**d**); Pq parameter of both profiles is 1 µm, sampling interval of both profiles is 2 µm.

**Figure 4 materials-16-07109-f004:**
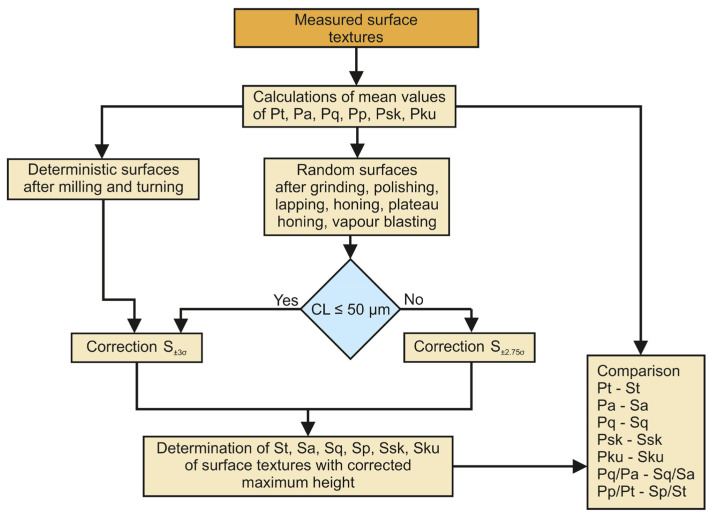
A scheme of analyses of measured surface textures.

**Figure 5 materials-16-07109-f005:**
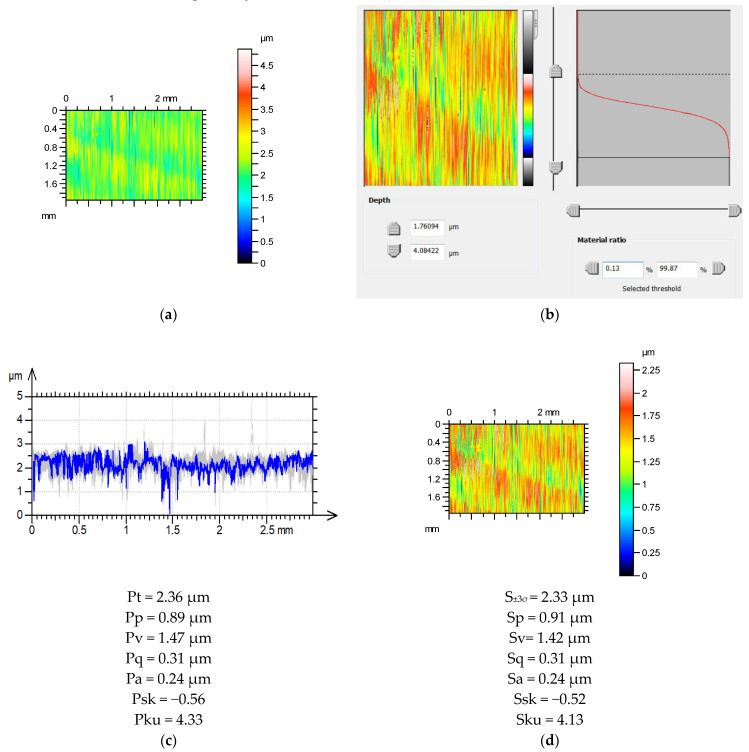
Estimations of profile parameters based on areal textures for surfaces after grinding (**a**–**d**) and vapor blasting (**e**–**h**); contour plots of measured surface textures (**a**,**e**), thresholding of the surfaces (**b**,**f**), series of profiles and average values of height parameters (**c**,**g**), contour plots of surface textures with corrected maximum height with height parameters (**d**,**h**).

**Figure 6 materials-16-07109-f006:**
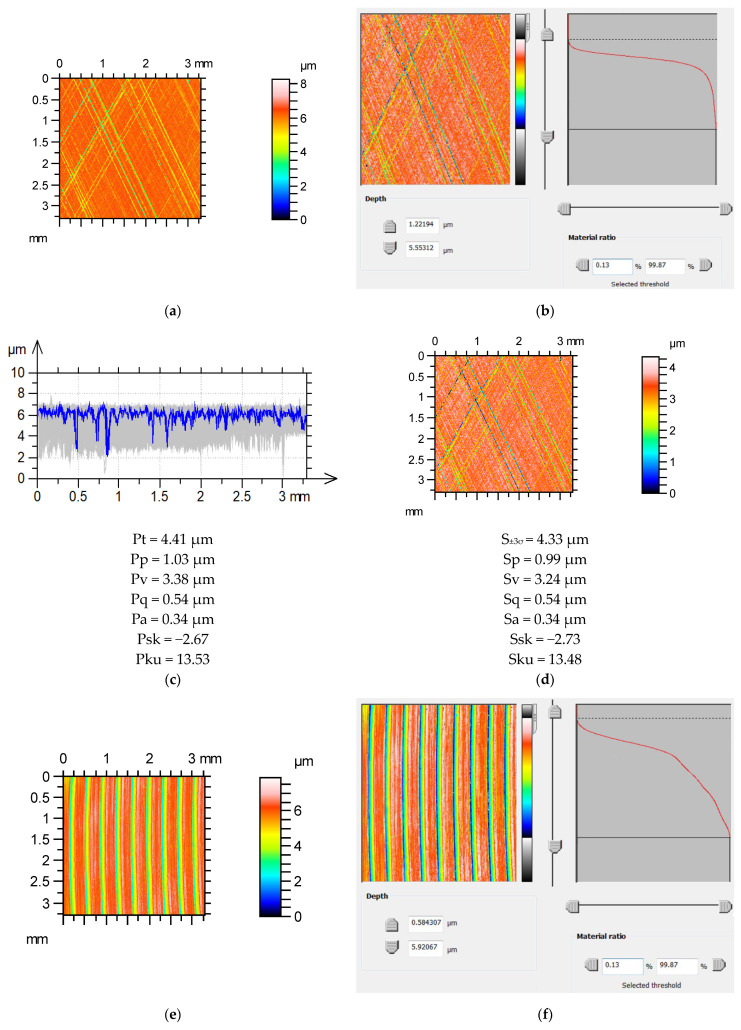
Estimations of profile parameters based on areal textures for surfaces after plateau honing (**a**–**d**) and milling (**e**–**h**); contour plots of measured surface textures (**a**,**e**), surfaces thresholding (**b**,**f**), series of profiles and average values of height parameters (**c**,**g**), contour plots of surface textures with corrected maximum height with height parameters (**d**,**h**).

**Table 1 materials-16-07109-t001:** Relative errors in estimations of mean values of selected profile parameters on the base of the analysis of areal surface textures.

Parameter	Mean Error, %	Maximum Error, %
Pt	3.3	9.7
Pq	1.3	6.7
Pa	1.0	9.3
Psk	73	6100
Pku	9.7	58.1
Pp/Pt	3.1	13.2
Pq/Pa	0.94	8.0

## Data Availability

Data are contained within the article.

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
