# Peer review of "Characterization of the Maximum Height of a Surface Texture"

_materials, 2023, doi:10.3390/ma16227109_

Round 1

Reviewer 1 Report

Comments and Suggestions for Authors

materials-2668962

Characterization of the maximum height of a surface texture

The content provided in this research article presents an intriguing method for estimating surface parameters from areal surface topography measurements. However, there are several areas that require significant revision and clarification to enhance the quality and comprehensibility of the work.

Comments:

  1. How does the choice of surface measurement method (stylus profilometer vs. white light interferometer) affect the accuracy of the estimated surface parameters?
  2. Can authors explain in more detail the correction process used to obtain stable parameters from areal surface topography measurements?
  3. What are the implications of the high variability in maximum profile height for industrial applications?
  4. How does the method perform when applied to surfaces with Gaussian ordinate distributions?
  5. Can authors provide examples of specific industrial applications where this method could be beneficial?
  6. What are the key limitations of the method when applied to surfaces with low correlation between neighboring ordinates?
  7. Are there any specific criteria for selecting between S±3σ and S±2.75σ correction methods for different surface types?
  8. How does the presence of spikes in optical measurements affect the accuracy of surface parameter estimation, and how is it mitigated?
  9. Can authors explain the significance of profile correlation lengths in relation to the method's accuracy?
  10. What are the challenges associated with measuring surfaces with very high or very low amplitude variations?
  11. How does the method perform when applied to surfaces after different manufacturing processes, such as grinding, polishing, and lapping?
  12. Are there any recommendations for improving the precision of the method in cases with negative skewness values?
  13. What are the practical implications of employing stitching procedures in surface topography measurements?
  14. How sensitive is the method to variations in the number of measuring points in perpendicular directions?
  15. Can authors provide insights into the sources of measurement errors in optical methods for surface topography?
  16. How does the method handle surfaces with bi-Gaussian random textures compared to Gaussian ones?
  17. What are the potential consequences of not correcting the maximum texture height to S±3σ for specific material ratios?
  18. Can authors elaborate on the significance of the horizontal distance between the 25 measuring points in relation to profile correlation lengths?
  19. Are there any real-world case studies or examples where this method has been successfully applied in industry?
  20. Can authors explain the practical implications of the mean error of the Pt parameter estimation being approximately 3%?
  21. How can this method be adapted or modified to accommodate longer correlation lengths in surface profiles?
  22. Are there any potential drawbacks or limitations to using S±3σ as an additional height parameter for areal surface texture?
  23. Can authors provide insights into the accuracy of the method when applied to very smooth surfaces?
  24. How do variations in surface parameters affect the performance of manufacturing processes?
  25. Are there any recommended best practices for selecting between different correction methods?
  26. Can authors explain the significance of profile amplitude parameters in industrial applications?
  27. How does the method perform when applied to surfaces with varying degrees of complexity?
  28. Can authors discuss the implications of the method's accuracy when applied to different types of manufacturing materials?
  29. Are there any plans or ongoing research to further improve the method's accuracy and applicability?
  30. Can authors provide guidance on how industries can implement this method effectively for quality control and process optimization?

Author Response

  1. How does the choice of surface measurement method (stylus profilometer vs. white light interferometer) affect the accuracy of the estimated surface parameters?

These two methods use different techniques; contact and optical, respectively. There are various sources of measurement errors after applications of these methods. They were utilized to prove that our approach can be used independently of measurement method.

2. Can authors explain in more detail the correction process used to obtain stable parameters from areal surface topography measurements?

Stable amplitude parameter was achieved by correction of maximum height.         The x direction corresponds to smaller correlation length in perpendicular directions of anisotropic surfaces. For random surfaces with correlation length CL not hgher  than 50 µm in the x direction  and for surfaces after turning and milling, the maximum amplitude was corrected to heights corresponding to material ratios between 0.13 and 99.87%. This corrected height was called S±3σ. When the correlation length of the random surfaces was greater than 50 µm, the maximum areal surface amplitude was corrected to a height corresponding to material ratios between 0.3 and 99.7%. This modified corrected height is denoted as S±2.75σ.

3. What are the implications of the high variability in maximum profile height for industrial applications?

Well-done surfaces should be falsely qualified (as shortages).

4. How does the method perform when applied to surfaces with Gaussian ordinate distributions?

This method performed well for Gaussian surfaces. Especially surfaces after polishing and lapping have Gaussian ordinate distributions.

 5. Can authors provide examples of specific industrial applications where this method could be beneficial?

Maximum and average surface heights are commonly used in most industrial applications. However, the maximum height fluctuates strongly between different surface measurement. The new parameter is more reproducible than maximum profile height calculated by usual manner.

6. What are the key limitations of the method when applied to surfaces with low correlation between neighboring ordinates?

This methods was restricted to surfaces, for which the profile correlation lengths were smaller than 75 µm or smaller than the horizontal distance between the 30 measuring points. For greater correlation lengths, further modification of procedure is needed. However, in scientific works, high correlation should be avoided.

7. Are there any specific criteria for selecting between S±3σ and S±2.75σ correction methods for different surface types?

For deterministic surfaces after milling and turning S±3σ should be applied. For other random surfaces this selection depends on the correlation length of some profiles (about five from one surface). When the profile correlation length was higher than 50 µm S±2.75σ should be used.

8. How does the presence of spikes in optical measurements affect the accuracy of surface parameter estimation, and how is it mitigated?

Presence of spikes causes overestimation and increase in variations of height parameters. The effect of spikes can be mitigated by various method such as correction of the maximum height  or limitation of slope.

 9. Can authors explain the significance of profile correlation lengths in relation to the method's accuracy?

Method of areal surface texture correction depends on profile correlation length. High correlation lengths can cause non-stable character of surface. In this case high-pass filtering will be necessary. However, we tested surfaces with low content of waviness.

10. What are the challenges associated with measuring surfaces with very high or very low amplitude variations?

Deterministic surfaces are characterized by lower amplitude variations than random surfaces. Homogeneous surfaces (with non-statistical peaks and valleys) had higher variability than inhomogeneous ones. It was  found that the corrected maximum height S±3σ was more stable than the maximum surface height St (or Sz).

11. How does the method perform when applied to surfaces after different manufacturing processes, such as grinding, polishing, and lapping?

The method performed better for surfaces after polishing and lapping compared to those after grinding. Most surfaces after lapping and polishing had Gaussian ordinate distribution which is probably related with good performance of the proposed method.

12. Are there any recommendations for improving the precision of the method in cases with negative skewness values?

This method performed well for bi-Gaussian surfaces after plateau honing. The performance of this method for other random surfaces with negative skewness values was poorer which is probably related with non-Gaussian character of these surfaces. The precision of this method can be improved by careful selection of the measured area.

13. What are the practical implications of employing stitching procedures in surface topography measurements?

  Stitching led to the possibility of increase in measurement area with good horizontal resolution.    

14. How sensitive is the method to variations in the number of measuring points in perpendicular directions?

When the number of measuring points in x direction is much smaller than 1000 and higher than 2000 the correction of the maximum height of areal texture should be modified.

15. Can authors provide insights into the sources of measurement errors in optical methods for surface topography?

This information has been added to Introduction section. Optical methods replaced stylus technique due to much shorter measurement time. However, optical methods are sensitive to measurement errors. Errors typical for optical methods are mainly caused by the presence of spikes and non-measured points. Spikes are narrow peaks that do not occur on a real surface. Non-measured points are surface locations for which no valid measured values occur. The application of stitching is an additional source of measurement error.

16. How does the method handle surfaces with bi-Gaussian random textures compared to Gaussian ones?

The results obtained for bi-Gaussian and Gaussian random surfaces were similar.

17. What are the potential consequences of not correcting the maximum texture height to S±3σ for specific material ratios?

Without correcting the maximum texture height one cannot estimate correctly maximum profile height and other profile parameters based on the parameters of areal surface texture.

18. Can authors elaborate on the significance of the horizontal distance between the 20 measuring points in relation to profile correlation lengths?

This value was consequence of the obtained results. In our research the sampling intervals were 2 and 3 µm and limitation of correlation length was 50 µm. Therefore this method should be used for horizontal distance between 50/2.5 = 20 measuring points. So, this method can be extended for other sampling intervals.

19. Are there any real-world case studies or examples where this method has been successfully applied in industry?

Truncation of surface texture height is the simplest method to remove spikes. However, in our opinion the limitation of slopes is better approach. The authors of papers [20, 2`] also corrected maximum surface height.

20. Can authors explain the practical implications of the mean error of the Pt parameter estimation being approximately 3%?

This  error of  Pt  estimation is low, compared with variations of profile parameters on typical manufactured surfaces.

21. How can this method be adapted or modified to accommodate longer correlation lengths in surface profiles?

The range of surface height should be narrowed – maximum surface height should be reduced.

22. Are there any potential drawbacks or limitations to using S±3σ as an additional height parameter for areal surface texture?

No, this parameter can be applied to all surfaces. If this parameter is utilized, impacts of non-statistical peaks or valleys will be minimized.

23. Can authors provide insights into the accuracy of the method when applied to very smooth surfaces?

Surfaces after polishing and lapping were very smooth. Accuracy of the Pt parameter estimation based was very good, mean error was 1.3% and maximum error was 3.1%.

24. How do variations in surface parameters affect the performance of manufacturing processes?

Surface control during production processes is easier when variations in surface parameters are lower.

25. Are there any recommended best practices for selecting between different correction methods?

The number of profiles (typically five was enough), for which the autocorrelation functions were determined depended on surface character; it should be higher for nonhomogeneous surfaces. The maximum heights of the areal surface textures were corrected based on the correlation lengths.

In industrial application correction S±2.75σ should be seldom used, because its necessity was found only for very rough surfaces; machined surfaces are typically smooth.

26. Can authors explain the significance of profile amplitude parameters in industrial applications?

The amplitude parameters are believed to mostly affect the surface properties for example they are related to friction and wear. The opinion exists that maximum height is related to surface damage while the averaged parameters to surface normal functioning. The maximum roughness height can be employed to detect cracks in the surface layer.

In industrial applications, profile analysis is most commonly utilized for the assessment of the machining process, and two amplitude parameters are typically applied: the arithmetical roughness mean height Ra and maximum heights Rz or Rt (Rz is calculated on the sampling length, whereas Rt is calculated on the evaluating length, the  sampling length is smaller). These parameters are employed to characterize the surface roughness in most engineering applications, because of their substantial functional importance and ease of use. 

27. How does the method perform when applied to surfaces with varying degrees of complexity?

Surface after plateau honing are complex surfaces.

Because this method performed well for plateau-honed surfaces and deterministic surfaces after milling and turning it would be probably useful for complex surfaces, such as textured surfaces.

28. Can authors discuss the implications of the method's accuracy when applied to different types of manufacturing materials?

This work focuses on surface texture analysis in the context of metals, however, the proposed method is probably applicable to other types of manufacturing materials, such as ceramic and composite.

29. Are there any plans or ongoing research to further improve the method's accuracy and applicability?

In further research the effect of high-frequency filtering on the accuracy of the proposed method will be studied. This method will be applied to other surfaces, including complex ones made from other materials.  

30. Can authors provide guidance on how industries can implement this method effectively for quality control and process optimization?

 The measurement of areal surface texture can be used in industry for production control and process optimization to obtain stable values of average and maximum height parameters. Based on surface type, the conditions of measurement and analysis such as measuring area, sampling interval should be determined. Filtering should be taken into account.

Reviewer 2 Report

Comments and Suggestions for Authors

1           This paper utilized both optical methods (Talysurf CCI Lite) and stylus methods (Talysurf i-series) to measure 100 surface conditions, covering a range of treatments including polishing, lapping, grinding, vapor blasting, turning, and milling. It's important to note, however, that the paper focuses exclusively on surface analysis in the context of metals (specifically stainless steel and gray cast iron). Could the methods proposed in this paper applicable to ceramic materials or composite materials?

2           The primary objective of this paper is to establish a parameter akin to the mean maximum height of parallel profiles. Additionally, it assesses the stability of Pp/Pt and Pq/Pa concerning Psk and Pku. The proposed method accurately estimates average values for parameters related to profile amplitude and amplitude distribution shape, including Pa, Pq, and Pq/Pa, with an average error of approximately 1%. It's important to note that numerous commercially available measurement devices already feature filtering and noise reduction capabilities, ensuring measurement stability. Consequently, the authors should provide further clarification regarding the novelty and practical application of this paper.

Author Response

  1. This paper utilized both optical methods (Talysurf CCI Lite) and stylus methods (Talysurf i-series) to measure 100 surface conditions, covering a range of treatments including polishing, lapping, grinding, vapor blasting, turning, and milling. It's important to note, however, that the paper focuses exclusively on surface analysis in the context of metals (specifically stainless steel and gray cast iron). Could the methods proposed in this paper applicable to ceramic materials or composite materials?

 This work focuses on surface texture analysis in the context of metals, however, the proposed method is probably applicable to other types of manufacturing materials, such as ceramic and composite.

  1. The primary objective of this paper is to establish a parameter akin to the mean maximum height of parallel profiles. Additionally, it assesses the stability of Pp/Pt and Pq/Pa concerning Psk and Pku. The proposed method accurately estimates average values for parameters related to profile amplitude and amplitude distribution shape, including Pa, Pq, and Pq/Pa, with an average error of approximately 1%. It's important to note that numerous commercially available measurement devices already feature filtering and noise reduction capabilities, ensuring measurement stability. Consequently, the authors should provide further clarification regarding the novelty and practical application of this paper.

In this work unfiltered parameters of corrected areal surface texture were compared with their substitutes of series of unfiltered profiles. However, in industry roughness profiles are commonly analysed. Therefore, before industrial application of this method, filtering which increases measurement stability,  should be taken into account. 

Reviewer 3 Report

Comments and Suggestions for Authors

The article deals with the possibilities of evaluating the machined surface, which is still a current topic suitable for the research of new approaches.
I have the following questions and comments for the authors on the presented article:

Abstract
is written clearly with the necessary background. What is missing is a better description of what the authors want to achieve with the research presented.

Introduction
is based on research of up-to-date information of authors from all over the world. 
However, the number of sources used is relatively small for such a challenging and complex topic.
Ra and Sa are still popular parameters, but only in mass production.

Chapters 2 and 3

Chapter 2, there is described the background to the calculation. The idea itself is generally applicable. However, for the use of surface control, it encounters specifics that are, among other things, described in the relevant standards and are not suitable for general application.
There is no explanation of how the equations can be applied directly to the surfaces being evaluated. What is the applicability? Is it possible to evaluate surfaces after turning and milling in this way? Rotating and non-rotating surfaces?
The authors state that a high number of samples were used (100).
Missing of samples characteraztion. Material, its mechanical properties, shape, dimensions, etc. Please add some figure (scheme) and detailed description.

Conclusions

It contains only a statement of the experiments carried out. The analysis and evaluation of the results are too brief.
What are the prospects for further research, outputs and implementations for practice?

Despite the undeniable efforts of the authors to present their own research in a high quality manner, I recommend major revision and only then look into the possibilities of publishing.

Author Response

  1. Abstract is written clearly with the necessary background. What is missing is a better description of what the authors want to achieve with the research presented.

Abstract has been improved.

  1. Introduction is based on research of up-to-date information of authors from all over the world. 
    However, the number of sources used is relatively small for such a challenging and complex topic.
    Ra and Sa are still popular parameters, but only in mass production.

The number of sources has been increased. Because Ra and Sa are important only in mass production, we considered the Sq parameter.

  1. Chapter 2, there is described the background to the calculation. The idea itself is generally applicable. However, for the use of surface control, it encounters specifics that are, among other things, described in the relevant standards and are not suitable for general application.
    There is no explanation of how the equations can be applied directly to the surfaces being evaluated. What is the applicability? Is it possible to evaluate surfaces after turning and milling in this way? Rotating and non-rotating surfaces?
    The authors state that a high number of samples were used (100).
    Missing of samples characteraztion. Material, its mechanical properties, shape, dimensions, etc. Please add some figure (scheme) and detailed description.

New information has been added. Equations (1) – (3) can be applied to random surfaces. Surfaces after turning and milling are deterministic surfaces. However, surface correction was used also for them to eliminate noise.

More information on the samples used has been added. Scheme of detailed description of the used procedure has been added.

  1. Conclusions

It contains only a statement of the experiments carried out. The analysis and evaluation of the results are too brief.
What are the prospects for further research, outputs and implementations for practice?

 Conclusions have been improved.

Round 2

Reviewer 1 Report

Comments and Suggestions for Authors

revised manuscript can be accepted

Reviewer 2 Report

Comments and Suggestions for Authors

The author has responded and revised the paper based on the reviewer's comments.

Reviewer 3 Report

Comments and Suggestions for Authors

I agree with the edits made and recommend the article for publication.